Relationship between fruit phenotypes and domestication in hexaploid populations of biribá (Annona mucosa) in Brazilian Amazonia

Serbin Giulia Melilli 1 giuliamelilli63@gmail.com
Pinangé Diego Sotero de Barros 2
Machado Raquel Moura 3
http://orcid.org/0000-0002-9717-1169 Vasconcelos Santelmo 4
Amorim Bruno Sampaio 5 6
http://orcid.org/0000-0002-8421-1029 Clement Charles Roland 3
1 Postgraduate Program in Botany, Instituto Nacional de Pesquisas da Amazônia , Manaus, Amazonas , Brazil
2 Department of Genetic, Universidade Federal do Amazonas , Manaus, Amazonas , Brazil
3 Instituto Nacional de Pesquisas da Amazônia , Manaus, Amazonas , Brazil
4 Instituto Tecnológico Vale , Belém, Pará , Brazil
5 Museu da Amazônia (MUSA) , Manaus, Amazonas , Brazil
6 Postgraduate Program in Biotechnology and Natural Resources of Amazonia, Universidade do Estado do Amazonas , Manaus, Amazonas , Brazil
Shrestha Jiban
Electronic publication date: 2023 Jan 23
Publication date: 2023
Volume: 11
Electronic Location ID: e14659
Received 2022 Oct 11; Accepted 2022 Dec 8
Copyright: © 2023 Serbin et al.
Copyright year: 2023
Copyright holder: Serbin et al.
License: This is an open access article distributed under the terms of the Creative Commons Attribution License, which permits unrestricted use, distribution, reproduction and adaptation in any medium and for any purpose provided that it is properly attributed. For attribution, the original author(s), title, publication source (PeerJ) and either DOI or URL of the article must be cited.
License URL: https://creativecommons.org/licenses/by/4.0/

Keywords: Amazonia, Biribá (Annona mucosa), Chromosome evolution, Domestication, Genome size, Polyploidy

Funding: Conselho Nacional de Desenvolvimento Científico e Tecnológico 435985/2018-3 Fundação de Amparo à Pesquisa do Estado do Amazonas 062.00148/2020 Instituto Tecnológico Vale (Belem, PA, Brazil) Coordenação de Aperfeiçoamento de Pessoal de Nível Superior 88882.444398/2019-01 Bruno Amorim postdoctoral fellowship by CAPES 88882.315044/2019-01 CNPq: 303477/2018-0 This work was supported by Conselho Nacional de Desenvolvimento Científico e Tecnológico (CNPq Universal; Process: 435985/2018-3) and the Fundação de Amparo à Pesquisa do Estado do Amazonas (FAPEAM; Process: 062.00148/2020). This work was supported by Instituto Tecnológico Vale (Belem, PA, Brazil) for the flow cytometry analyses and for financial support for publication. Giulia Melilli Serbin Master’s scholarship was supported by Coordenação de Aperfeiçoamento de Pessoal de Nível Superior (CAPES; Process: 88882.444398/2019-01); Bruno Amorim postdoctoral fellowship by CAPES (Process: 88882.315044/2019-01); Charles Clement research fellowship by CNPq (Process: 303477/2018-0). The funders had no role in study design, data collection and analysis, decision to publish, or preparation of the manuscript.

==============================
Background

Biribá (Annona mucosa Jacq.) is a fruit tree domesticated in Amazonia and has polyploid populations. The species presents ample phenotypic variation in fruit characteristics, including weight (100–4,000 g) and differences in carpel protrusions. Two cytotypes are recorded in the literature (2n = 28, 42) and genome size records are divergent (2C = 4.77, 5.42 and 6.00 pg). To decipher the role of polyploidy in the domestication of A. mucosa, we examined the relationships among phenotypic variation, chromosome number and genome size, and which came first, polyploidization or domestication.

Methodology

We performed chromosome counts of A. mucosa from central and western Brazilian Amazonia, and estimated genome size by flow cytometry. We performed phylogenetic reconstruction with publicly available data using a Bayesian framework, time divergence analysis and reconstructed the ancestral chromosome number for the genus Annona and for A. mucosa.

Results

We observed that variation in fruit phenotypes is not associated with variation in chromosome number and genome size. The most recent common ancestor of A. mucosa is inferred to be polyploid and diverged before domestication.

Conclusions

We conclude that, when domesticated, A. mucosa was already polyploid and we suggest that human selection is the main evolutionary force behind fruit size and fruit morphological variation in Annona mucosa.

Introduction

The gradual transition from hunting and gathering to growing plants began before 12,000 years ago, between the late Pleistocene and early Holocene (Fuller et al., 2014), and involved both domestication of crops and development of food production systems. Domestication results in a set of modified traits called the domestication syndrome, that differentiates them from their wild ancestors (Pickersgill, 2007; Meyer, Duval & Jensen, 2012); an example is the increase in size of useful parts of the plant (Hancock, 2012). About 30% of cultivated plants are polyploid (Salman-Minkov, Sabath & Mayrose, 2016), which refers to the multiplication of a complete chromosome set (Soltis et al., 2015; Zhang, Wang & Cheng, 2019). Polyploidization can occur before or after domestication, generating phenotypic changes that are often of human interest and concentrated in the domestication syndrome traits (Hancock, 2012; Zhang, Wang & Cheng, 2019). Domestication is widespread in the Americas (Meyer, Duval & Jensen, 2012; Clement et al., 2021), and in Amazonia at least 85 tree and palm species present populations with some degree of domestication (Levis et al., 2017). However, information about polyploidy in Amazonian crops is limited (Rice et al., 2019), which makes biribá (Annona mucosa Jacq.), a popular fruit crop, a good case study.

The Annonaceae includes 107 genera (Guo et al., 2017), with 2,400 species, many of which occur in Amazonia (Maas, Lobão & Rainer, 2015; Guo et al., 2017). The family originated about 99 million years ago (Mya) and most species diversified before 1.9 Mya (Li, Thomas & Saunders, 2017). Within the family, the genus Annona stands out for presenting many species with edible fruit in different stages of domestication (Table S1), such as cherimoya (Annona cherimola Mill.), sugar apple (Annona squamosa L.), soursop (Annona muricata L.), atemoya (a hybrid between A. cherimola and A. squamosa), custard apple (Annona reticulata L.), ilama (Annona macroprophyllata Donn. Sm.), pond apple (Annona glabra L.), soncoya (Annona purpurea Moc. & Sessé ex Dunal), and biribá (A. mucosa) (Larranaga, Albertazzi & Hormaza, 2019; Araujo et al., 2021). Currently, the genus has ca. 160 species, with a Pantropical distribution, forming a monophyletic group, that recent modelling shows diverged 52.5 Mya (Li, Thomas & Saunders, 2017; Maas et al., 2019). The monophyly of Annona depended on the inclusion of the genus Rollinia (Rainer, 2007; Chatrou et al., 2009), whose flowers differ from those of other species of Annona in that they have a gammopetal corolla composed of three large external lobes and three alternate smaller internal lobes (Maas et al., 1992). The species that belonged to the genus Rollinia now form the Rollinia clade (Chatrou et al., 2009), which includes A. mucosa and 42 other species (Rainer, 2007) with an exclusively Neotropical distribution (Maas et al., 1992).

Annona mucosa is a fruit tree cultivated by pre-Columbian peoples and modern communities (Clement, 1999; Clement et al., 2021), and is used principally as a fresh fruit, although it also has medicinal properties (Cavalcante, 2010). The distribution of cultivated A. mucosa is ample, from Mesoamerica to southern Brazil (Maas et al., 1992). Wild populations have been reported in Peruvian Amazonia (Patiño, 2002), the state of Rondônia in Brazil (Cavalcante, 2010) and recently in Mexico (Segura et al., 2018; Escobedo-López et al., 2019). In Brazil, the species occurs more frequently in Amazonia (Maas, Lobão & Rainer, 2015), where fruit morphological variation is significant (Clement, 1989). The fruit is a syncarp and domesticated populations present different carpel protrusions, as well as different sizes (from 100 to 4,000 g). The largest fruits are found in the upper Solimões River region of western Amazonia (Clement, 1989; Cavalcante, 2010).

The morphological variations found in Amazonian biribás may reflect human selection, polyploidy or both. Like domestication, polyploidy generates genomic changes (Soltis et al., 2015). Due to the larger number of chromosomes, polyploidy offers a greater chance of gene recombination, directly affecting gene expression (Grant, 1974; Guerra, 1988; Hancock, 2012). There is often a correlation between fruit size and increasing ploidy level in fruit crops, such as strawberry (Fragaria × ananassa Duch.) (Bertioli, 2019) and star fruit (Averrhoa carambola L.) (Hu et al., 2021). However, it is often unknown whether a given population was selected for being polyploid or whether polyploidy was established after selection began.

The genus Annona has a basic chromosome number of x = 7 (Morawetz, 1986a, 1986b). Annona mucosa has two ploidy levels: 2n = 4x = 28 in a population in Peru (Maas et al., 1992) and 2n = 6x = 42 in domesticated populations in Brazil (Morawetz, 1986b; Lorenzoni, 2016). The species also appears to present genome size variation (2C = 4.77, 5.42 and 6.00 pg) (Soares et al., 2014; Lorenzoni, 2016; Leitch et al., 2019). However, the cytogenetic and genome size studies of A. mucosa have many gaps, such as the absence of exact collection location and the lack of information about the phenotype of the fruits in the published studies.

The different genome sizes presented in the literature could be associated with different chromosome counts, establishing a direct relationship between genome size and cytotypes (Machado et al., 2021). In the absence of cytotypes, intraspecific variation in genome size could be due either to fluctuations within highly repetitive DNA or to structural rearrangements (Bennetzen, Ma & Devos, 2005; Šmarda & Bureš, 2010). In the case of structural reorganization, it can affect expression patterns that affect the phenotype, such as increased biomass and morphotypes (Balao, Herrera & Talavera, 2011; Meyerson et al., 2020). As phenotypic variation has not yet been associated with different cytotypes and genome sizes in A. mucosa, studying this relationship can provide answers about the role of polyploidy in the origin of phenotypic diversity.

Annona does not have a high rate of polyploidization, although the Rollinia clade does (Morawetz, 1984; Maas et al., 1992). The high frequency of polyploids within a clade may be associated with ancestral polyploidy events (Rice et al., 2019; Sader et al., 2019). Phylogenetic studies make it possible to understand the evolutionary relationships among species and infer the timing and mode of divergence events (Sader et al., 2019) and their relationship to cariological traits (Glick & Mayrose, 2014; Salman-Minkov, Sabath & Mayrose, 2016). These analyses allow inferences about ancestral polyploidy events that could explain the high frequency of polyploids in a lineage and establish the spatiotemporal relationship between domestication and polyploidy (Moraes et al., 2020; Urdampilleta, Forni-Martins & Ferrucci, 2020).

Given this context, we consider A. mucosa as a model to analyze the role of polyploidy in domestication processes in Amazonia. Specifically, we test: (1) whether the phenotypic variation in A. mucosa fruits (sizes and morphotypes) is associated with different chromosome numbers and genome sizes; and (2) whether the polyploidization event(s) occurred before the domestication of the cultivated populations.

Materials and Methods

Plant material

The fruits were purchased at fairs in central Amazonia, in the state of Amazonas (Manaus and Rio Preto da Eva), along the middle (Tefé) and upper Solimões River (Benjamin Constant, Tabatinga and Atalaia do Norte) in western Amazonia (Table 1 and Fig. 1A) (SISBIO authorization 70846-1 and SisGen registration A19391B). We classified the fruits found into five different morphotypes based on the carpel protrusions on the exocarp (Fig. 1B), as done for cherimoya (A. cherimola) (Vanhove, 2008): smooth fruits; fruits with small carpel depressions; fruits with small carpel protrusions; fruits with medium carpel protrusions and fruits with large carpel protrusions.

Table 1 Fruits of Annona mucosa Jacq. used in the study.

N	City	Weight	2n	CMA+	2C	CV	Morphotypes	Voucher	
1	Manaus	100	6x = 42	5	–		Small	–	
2	Rio Preto da Eva	165	6x = 42	5	5.32 (0.13)	3.73	Small	287053 (INPA)	
3	Rio Preto da Eva	235	6x = 42	5	5.30 (0.06)*	2.94	Smooth	287054 (INPA)	
4	Manaus	300	6x = 42	5	5.23 (0.02)	3.23	Small	–	
5	Manaus	350	6x = 42	5	–		Small	–	
6	Manaus	350, 450, 550, 650, 750, 1,050	6x = 42	5	–		Small	287047 (INPA)	
7	Manaus	350	6x = 42	–	–	–	Medium	–	
8	Manaus	400	6x = 42	5	–	–	Small	–	
9	Tabatinga	400	–	–	5.41 (0.07)	3.26	Large	287902 (INPA)	
10	Manaus	450	6x = 42	5	–	–	Medium	–	
11	Manaus	450	6x = 42	5	–	–	Small	–	
12	Manaus	450, 700	6x = 42	5	5.26 (0.08)	2.94	Small	287048 (INPA)	
13	Manaus	450	6x = 42	5	5.23 (0.04)	3.24	Small	Living collection	
14	Tefé	500	6x = 42	5	5.32 (0.06)	3.37	Small depression	Living collection	
15	Tefé	500	6x = 42	5	–		Small	–	
16	Tefé	500	6x = 42	5	5.65 (0.07)	3.72	Medium	Living collection	
17	Benjamin Constant	650	6x = 42	5	5.28 (0.23)	3.42	Large	Living collection	
18	Manaus	700	6x = 42	5	–	–	Medium	–	
19	Manaus	700	6x = 42	5	–	–	Small	–	
20	Manaus	700	6x = 42	5	5.28 (0.05)	3.25	Medium	Living collection	
21	Manaus	1,020	6x = 42	–	–		Large	287049 (INPA)	
22	Manaus	1,200	6x = 42	5	5.31 (0.03)	3.15	Large	Living collection	
23	Tabatinga	1,200	6x = 42	5	5.33 (0.07)	3.19	Large	Living collection	
24	Tabatinga	1,300	6x = 42	5	5.27 (0.12)	3.33	Large	Living collection	
25	Benjamin Constant	1,600	6x = 42	–	5.21 (0.06)	3.19	Medium	Living collection	
26	Atalaia do Norte	1,850	6x = 42	5	5.26 (0.06)	3.49	Large	Living collection	
Notes:

Fruits of Annona mucosa Jacq. used in the study, with the city of collection in the state of Amazonas, Brazil, fruit weight (grams), chromosome number (2n), numbers of CMA+ band pairs, genome size (2C in picograms, mean (standard deviation)), coefficient of variation (CV) of genome size, morphotype based on carpel protrusions (Morphotypes; Vanhove (2008)), and the collection voucher (Herbarium at INPA).

* Measurements of DNA content by flow cytometry were performed with only two runs. Living collections in the Estação Experimental de Fruticultura Tropical, Instituto Nacional de Pesquisas da Amazônia, Manaus, Amazonas, Brazil.

Figure 1 Distribution of the morphotypes of Annona mucosa used in this study.

(A) Collection locations in central Amazonia (Manaus and Rio Preto da Eva), and along the middle (Tefé) and upper Solimões River (Benjamin Constant, Tabatinga and Atalaia do Norte), Amazonas, Brazil; (B) morphotypes, size range of the fruits in grams (g) and the localities where they were collected, the color of the circle is associated with the regions on the map (A).

Chromosome counts

Chromosomal preparations were performed according to Guerra & Souza (2002), with some adaptations for A. mucosa. The seeds were germinated, and the root apices pretreated with 8-hydroxyquinoline (8 HQ) 2 mM for 24 h in a refrigerator. Roots were fixed in Carnoy (ethyl alcohol: acetic acid, 3:1) for about 6 h at room temperature (~25 °C) and then stored in a freezer (−20 °C). Slides were prepared with enzymatic digestion (1% macrozyme, 2% cellulase and 20% pectinase) for 40 min at 37 °C.

After 5 days, we used the Schweizer (1976) protocol to band the chromosomes in the slides, with the following modifications. Slides were covered with 12 µL of chromomycin A3 (CMA; 0.5 mg/ml) in a darkroom for 1 h, then washed with distilled water and covered with 12 µL of 4–6-diamidino-2-phenylindole (DAPI; 2 mg/ml) in a dark chamber for 30 min. Then, slides were mounted with 12 µL of glycerol/McIlvaine/MgCl2 and covered with a glass coverslip. The slides were then stored for at least three days until observation under an Olympus BX51 microscope. Images were captured using the Olympus Cell View program (Olympus Corporation, Shinjuku City, Japan). Adobe Photoshop CS6 (Adobe Systems, Inc., San Jose, CA, USA) was used to assist in balancing colors, contrast and brightness, and assembling the chromosome image.

Genome size estimations by flow cytometry

Young leaves of 15 A. mucosa individuals were collected and used for genome size measurements by flow cytometry. Approximately 1 cm2 of young leaf tissue was minced with fresh leaf material from an internal standard (Petroselinum crispum (Mill.), Apiaceae, 2C = 4.50 pg). Nuclear suspensions were prepared according to Doležel, Greilhuber & Suda (2007), adding 1.5 mL of WPB (woody plant buffer) (Loureiro et al., 2007) to the nucleus extract, and then the nuclear suspension was filtered through a 30 μm nylon mesh. The nuclear suspension was stained with 50 μL of 1 mg/mL propidium iodide, incubated for at least 10 min and analyzed by flow cytometry. At least 5,000 nuclei from three replicates were analyzed for each sample using a CyFlow Ploidy Analyzer (Sysmex, Kobe, Japan) cytometer. Each set of histograms from the flow cytometry analyses was analyzed using Flowing Software v2.5.1 by Perttu Terho (Turku Center for Biotechnology, University of Turku, Turku, Finland).

Statistical relationship between fruit weight, morphotypes and genome size

To test whether the samples originated from the same distribution and to analyze the relationship among genome size, morphotype and fruit weight of A. mucosa individuals (non-normal data), we performed the Kruskal-Wallis test (McKight & Najab, 2010), a nonparametric ANOVA, in R package MultNonParam (Kolassa & Jankowski, 2014).

Sequence editing, alignment and phylogenetic analysis

We sampled 50 taxa of the genus Annona available in GenBank (Benson et al., 2012), which represents 31% of the accepted species of the genus (Table S2). Among the 50 taxa of Annona, 16 were from the Rollinia clade (38% of the 42 species in Rollinia). The species Asimina incana (W. Bartram) Exell. was used as an outgroup following Li, Thomas & Saunders (2017). The plastid intergenic spacer psbA-trnH and the plastid genes matK, ndhF, rbcL and trnL were used, totaling 161 sequences (Table S3). The DNA sequences were aligned using MUSCLE (Edgar, 2004) as an extension of Geneious v.7.1.9 (Kearse et al., 2012). Subsequently, the sequences of markers for each of the species were concatenated for analysis, as done by Guo et al. (2017) in the major phylogeny for the Annonaceae.

We used jModelTest v.2.1.6 to assess the best DNA substitution model for each individual marker (Darriba et al., 2012) evaluated with the Akaike Information Criterion (Akaike, 1974). The best fitting models were: rbcL (K80+I), trnL (HKY+I), matK (TPM1uF+G), ndhF (HKY+I), psbA-trnH (HKY+G) (Table S3). Phylogenetic relationships were inferred using Bayesian Inference with four independent runs with four Markov Chain Monte Carlo (MCMC) runs, sampling every 1,000 generations for 10,000,000 generations, implemented in MrBayes v.3.2.6 (Ronquist et al., 2012). The analysis was evaluated in TRACER v.1.6 (Rambaut & Drummond, 2014) to determine if the estimated sample size (ESS) was greater than 200, after applying a burn-in of 25%. The consensus tree was viewed and edited in FigTree v.1.4.2. (Rambaut, 2009). All phylogenetic analyses, excluding the ESS, were performed on the CIPRES Science Gateway (Miller, Pfeiffer & Schwartz, 2011).

Estimates of divergence time using a molecular clock

For the 22 taxa with chromosomal information (Table S1), we repeated the phylogenetic analyses following the same methodology described above. We estimated the divergence time between species in BEAST v.1.8.3., with input preparation by BEAUti (Drummond & Rambaut, 2007; Drummond et al., 2012). We used the “Relaxed Clock: Uncorrelated Lognormal” (Drummond & Rambaut, 2007) and the “Yule” specification model (Gernhard, 2008). Two independent runs of 200,000,000 generations were performed, sampled every 10,000 generations. To verify the effective sampling of parameters and to assess the convergence of the independent chains, we examined the posterior distributions in TRACER v.1.6. (Rambaut & Drummond, 2014). Markov Chain Monte Carlo (MCMC) sampling was considered sufficient at effective sample sizes (ESS) greater than 200. After removing 10,000 samples as burn-in, independent runs were combined, and a maximum clade credibility tree (MCC) was constructed using TreeAnnotator v.1.8.2. (Drummond et al., 2012). BEAST analyses were performed on the CIPRES Science Gateway (Miller, Pfeiffer & Schwartz, 2011). The tree calibration used secondary calibrations, based on the work of Li, Thomas & Saunders (2017); for the node of the monophyletic genus Annona (29 Mya) and the node of the group Annonae (53.3 Mya) a standard deviation of two was used for both.

Reconstruction of ancestral character state

To reconstruct the ancestral ploidy state, we obtained chromosome numbers (n) of Annona species from The Chromosome Counts Database (Rice et al., 2015) and the Index to Plant Chromosome Numbers (Goldblatt & Johnson, 1994). The genome size data (1C) were taken from the Plant DNA C-values Database (Leitch et al., 2019) and literature (Table S1).

For the 22 species with chromosome number information (Table S1), the evolution of haploid chromosome numbers was inferred using two approaches. The first analysis was performed using ChromEvol v.2.0 (Glick & Mayrose, 2014). This program determines the likelihood of a model explaining the given data within the phylogeny based on the combination of two or more parameters. Ten models were tested with different sets of program parameters. The models were fitted to the data, each with 10,000 simulations, and the best fit model was Base_Number_No_Dupl selected using the AIC (Table S4). In addition to the AIC estimates, we tested the model suitability for probabilistic models of chromosome number evolution according to Rice & Mayrose (2021). The second analysis was performed using R package phytools (Revell, 2012), considering the haploid number of species (n = 7, 14, 21 and 28). The characters were reconstructed using function “ace” of the R package Ape (Paradis, Claude & Strimmer, 2004) with the EqualRates model. In cases where multiple chromosome numbers were reported for a given taxon, the model number was used (Glick & Mayrose, 2014; Salman-Minkov, Sabath & Mayrose, 2016).

Results

Fruit morphometrics and karyotypes

The weights of A. mucosa fruits ranged between 100–1,850 g (Table 1), with the largest fruits from the upper Solimões River region in western Amazonia. The morphotypes based on the classification of carpel protrusions were variable. The most frequent morphotype in the Manaus region had small carpel protrusions (58%), while in the upper Solimões River region the most frequent morphotype had large carpel protrusions (89%). Smooth and small depression morphotypes were rare (Table 1, Figs. 1A and 1B). In the Kruskal-Wallis analysis, there was no significant relationship between fruit size and morphotype, only a tendency for larger protrusions to occur in larger fruits (p = 0.0646).

All the 31 fruits that were karyotyped had the same chromosome number of 2n = 6x = 42 (Table 1 and Fig. 2). The distribution of constitutive heterochromatin (HC) was analyzed quantitatively. Five pairs of bands with a strong signal of CMA+/DAPI− were observed, three in regions close to the centromere (pericentromeric) and two in the short arms of chromosomes (with distension). There was no discernable relationship among the sizes, morphotypes and HC distributions.

Figure 2 Annona mucosa morphotypes and their respective metaphase cells.

(A) Smooth fruits; (E) fruits with small carpel depressions; (I) fruits with small carpel protrusions; (M) fruits with medium carpel protrusions; (Q) fruits with large carpel protrusions; (B, F, J, N and R) metaphases stained with DAPI; (C, G, K, O and S) metaphases stained with CMA; (D, H, L, P and R) overlayed CMA-DAPI images. Bar 20 µm.

The genome sizes of the 15 A. mucosa individuals analyzed revealed little variation and no significant differences between individuals (2C mean = 5.31 ± 0.12, minimum = 5.21 pg, maximum = 5.65 pg) (Table 1 and Fig. S1). The genome size was not related to fruit size (p = 0.2267) or morphotype variation (p = 0.6437).

Phylogenetic relationships, diversification rate and polyploid state reconstructions in Annona

With the 50 species of Annona examined with five molecular markers (Table S3), the genus was recovered as monophyletic (PP 100%), as was the Rollinia clade (PP 96%) (Fig. S2). The tree topology shows that A. mucosa and A. cuspidata ((Mart.) H. Rainer) shared the same most recent common ancestor (MRCA) (PP 98%) (Fig. S2).

With the 22 species of Annona for which both DNA sequences and chromosome counts were available, the tree topology was similar (Figs. 3A and 3B; Fig. S3) and A. mucosa and A. cuspidata also shared the same MRCA (PP 93%). The BEAST analysis estimated that the divergence of genus Annona occurred in the Paleocene (59.2 Mya: 95% CI [58.3–60.2]), but its diversification appears to have started in the Oligocene when the three largest groups diverged (29 Mya: 95% CI [28–30]). The BEAST analysis estimated that during the Oligocene the Rollinia clade began to diversify (23 Mya: 95% CI [17.8–28.4]), and A. mucosa diverged in the Pliocene (3.4 Mya: 95% CI [0.3–9.2]).

Figure 3 Reconstruction of ancestral haploid chromosome number (n) of Annona species.

(A) By ChromEvol and (B) with R package Phytools. The small circles beside the names of the species represent the haploid chromosome number (n); the large circles at the phylogeny nodes contain the probable haploid chromosome number of the ancestor; the timeline was extracted from the maximum clade credibility tree (MCC) from the BEAST analysis and the nodes represent later ages (Millions of years ago—Mya). P, Present; Ple, Pleistocene; Pli, Pliocene; Pal, Paleocene.

The reconstruction using ChromEvol revealed x = 7 as the ancestral chromosome number for the genus Annona (Fig. 3A), as did the reconstruction using the R package Phytools (Fig. 3B). With the model of Chromevol, we found one independent genome duplication event in the MRAC of A. mucosa and A. cuspidata, i.e., x = 14. With the model of Phytools, we found two independent genome duplication events in the MRAC of A. mucosa and A. cuspidata, with similar probabilities, i.e., x = 14 and x = 21. The probability of a polyploid ancestor for the Rollinia clade is low in both approaches.

Discussion

Domesticated A. mucosa has hexaploid populations in Brazilian Amazonia. The ample phenotypic variation in fruit characteristics, including weight and differences in carpel protrusions, is not associated with ploidy level or with different genome sizes. The phylogenetic analyses suggest that polyploidy occurred before domestication, as the most recent common ancestor of A. mucosa diverged in the Pliocene (well before humans arrived in the Americas) and was inferred to be polyploid.

Evolutionary events and their consequences for the phenotypic variation of A. mucosa

A primary potential advantage of polyploidy for domestication of fruit crops is fruit size variation (Salman-Minkov, Sabath & Mayrose, 2016; Akagi et al., 2022). Since polyploidy is associated with gigantism of reproductive parts in plants (Hancock, 2012), the ploidy level was expected to be associated with different fruit sizes, i.e., smaller fruits should present a smaller ploidy level and as the fruit size increases, the ploidy level would increase proportionally. This difference in ploidy may also represent different stages of domestication or different domestication syndromes in species of a genus (Akagi et al., 2022). This association has been observed in some fruit crops, for example, in the domestication of strawberry (Bertioli, 2019) and star fruit (Hu et al., 2021). However, this hypothesis was not confirmed for Annona mucosa. Within Annona, plants with very large fruits, such as domesticated soursop (A. muricata), are diploid, and plants with small fruits, such as undomesticated Araticum do mato (Annona sylvatica A. St.-Hil.), are octaploid (Table S1), demonstrating that wild species have variable ploidy levels also. Thus, we infer that human selection in Annona mucosa influenced fruit size regardless of the ploidy level.

The other hypothesis raised concerns the presence of variation in genome size related to different phenotypes, which was also refuted. Even in different regions of Brazilian Amazonia, the variation in genome size was not significant, suggesting stability in the genome. The genome size reported here is similar to Lorenzoni (2016): 2C = 5.42 pg (±0.12) with 15 accessions of unknown origin of hexaploid cultivated plants maintained at the Federal University of Espírito Santo (Alegre, ES, Brazil). The other two studies showed divergences in the size of the A. mucosa genome (Soares et al., 2014; Leitch et al., 2019), but methods, sample sizes and provenances are unclear.

Martin et al. (2019) established a DNA amount for diploid individuals in Annona and associated increased ploidy with genome size. Compared to A. neosalicifolia H. Rainer, a hexaploid species (2C = 9.64 pg), A. mucosa has approximately four pg less DNA content. Based on the genome sizes of other Annona species (Table S1), we observed that for A. mucosa cultivated in Brazilian Amazonia this characteristic is not linearly associated with ploidy, as all samples have 6x = 42 and 2C = 5.31 ± 0.12. Genome size reduction is a recurrent phenomenon in polyploid plants, which could explain this reduction in A. mucosa (Bennetzen, Ma & Devos, 2005; Šmarda & Bureš, 2010). The reduced 2C mean value in higher ploidy levels may be due to the loss of repetitive DNA, as in Psidium cattleyanum Sabine (Myrtaceae), a neotropical polyploid complex frequently managed by humans (Machado & Forni-Martins, 2022).

The species also does not follow the pattern of heterochromatic regions of the genus Annona, presenting five pairs of bands, while the expected would be three pairs, based on diploid and tetraploid species (Morawetz, 1986b). In polyploidy, the larger number of chromosomes offer a greater chance of gene gain and recombination (Grant, 1974; Guerra, 1988; Hancock, 2012), which could explain the difference in the pattern of the heterochromatic regions.

Polyploidization events and the evolutionary context of the domestication of A. mucosa

Despite the under representation of our plastidial phylogenetic hypothesis (with 50 species in general and 22 species for chromosome number reconstruction), our data corroborate the existing phylogenies published for Annona (Chatrou et al., 2009; Guo et al., 2017). This issue of lower representation is due to the limitation of available ploidy data, which is relatively common in studies involving compilation of molecular data and chromosomal counts or genome size (Sader et al., 2019; Ibiapino et al., 2022). Even though our results recovered all the main clades within Annona with high statistical support, our lower taxon representation may have favored this high internal resolution because of the increasing ratio of molecular data to the number of terminals, as already mentioned for mega diverse neotropical genera (Amorim et al., 2019). As well as the phylogenetic relationships, our divergence time estimations agree with what is known for the genus Annona and the divergence of A. mucosa and A. herzogii in the Pleistocene (Li, Thomas & Saunders, 2017).

The species of the Rollinia clade, which had previously presented polytomy (Chatrou et al., 2009; Guo et al., 2017), had good statistical support for the branches in our analysis. This clade resolution allowed us to suggest that A. mucosa and A. cuspidata share the same MRCA given our sample of 38% of the species in the Rollinia clade (Fig. S2). The ancestral chromosome number reconstructions presented in our study are divergent in relation to the polyploid ancestry of the Rollinia clade. Similar divergences have been reported in other plant groups and, depending on the methodology, different chromosomal events are suggested (Brottier et al., 2018; Moraes et al., 2020). With our results it is not possible to infer whether polyploidy is a consequence of a single or multiple ancestral polyploidization events within the Rollinia clade because we sampled only 16% of the clade’s species with chromosome data in the reconstruction (Figs. 3A and 3B).

On the other hand, our results suggest (in both approaches) that the MRCA of A. mucosa was polyploid. Because this ancestor was estimated to have diverged in the Pliocene, before human arrival in the Americas (Clement et al., 2021), the polyploidy event is inferred to have occurred well before the domestication process began. This temporality of events has already been observed in guarana (Paullinia cupana Kunth var. sorbilis Ducke), a species of vine domesticated in central Amazonia (Urdampilleta, Forni-Martins & Ferrucci, 2020), and in numerous other plant species (Salman-Minkov, Sabath & Mayrose, 2016; Zhang, Wang & Cheng, 2019).

Because of our inference about the polyploid ancestor of A. mucosa, it is expected that wild populations of A. mucosa will be polyploid. Three individuals in possibly non-anthropic environments in Peru were recorded as tetraploid (Maas et al., 1992), but only one sample was described as collected in primary forest. Additionally, an individual in an apparently wild population of A. mucosa in Veracruz, Mexico, presented hexaploidy (Serbin et al., 2022). This result reinforces the hypothesis of the presence of polyploidy before the domestication process and that polyploidy was not the main evolutionary force driving human selection of A. mucosa.

Conclusions

We conclude that the ample phenotypic variation in fruit characteristics is not associated with an increase in ploidy level or with different genome sizes in Annona mucosa domesticated in Brazilian Amazonia, suggesting that human selection is the main evolutionary force behind fruit size and morphological variation. We infer that polyploidy occurred before domestication, as the most recent common ancestor of A. mucosa diverged in the Pliocene (well before humans arrived in the Americas) and was also inferred to be polyploid. Thus, new questions arise, such as, what is the origin of the hexaploidy of domesticated and wild A. mucosa? Only more intensive collecting and karyotyping will answer this question.

Supplemental Information

Supplemental Information 1 Fluorescence histograms of Annona mucosa..

Fluorescence histograms of simultaneous analysis of propidium iodide-stained nuclei isolated from fresh tissue of internal standard Petroselinum crispum (Mill.) and Annona mucosa, respectively.

Click here for additional data file.

Supplemental Information 2 Phylogenetic reconstruction of the genus Annona.

The tree was based on plastid sequences: rbcL, matK, psbA-trnH, trnL and nadH from Genebank. The number in front of the nodes represents the posterior probability of the Bayesian analysis.

Click here for additional data file.

Supplemental Information 3 Chronogram of Annona based on rbcL, matK, psbA-trnH, trnL and nadH.

The timeline was extracted from the maximum clade credibility tree (MCC) of the BEAST analysis. Nodes represent later mean ages (Millions of years ago—Mya). The blue bars at the nodes represent the highest posterior density ranges with 95% Confidence Intervals. Numbers at nodes represent Bayesian posterior probabilities (PP).

Click here for additional data file.

Supplemental Information 4 Chromosome number, genome size and degrees of domestication available for the genus Annona..

Missing data (-).

Click here for additional data file.

Supplemental Information 5 Species names and GenBank accession numbers of DNA sequences used in this study.

Missing data (-).

Click here for additional data file.

Supplemental Information 6 Statistics of the markers used in the phylogenetic analysis of 50 species of Annona..

For each marker, the number of species with information, the number of characters aligned in base pairs (bp), the percentage of conserved characters and the evolutionary substitution model are presented.

Click here for additional data file.

Supplemental Information 7 Log-likelihood and Akaike information criterion (AIC) score estimates for the dataset analyzed by the ChromEvol.

Click here for additional data file.

The authors thank Professors Eliana Feldberg, Eliana Regina Forni Martins and Doriane Picanço Rodrigues for the availability of their laboratories for the cytogenetic analyses.

Additional Information and Declarations

Competing Interests

Author Contributions

Data Availability

The authors declare that they have no competing interests.

Giulia Melilli Serbi conceived and designed the experiments, performed the experiments, analyzed the data, prepared figures and/or tables, authored or reviewed drafts of the article, and approved the final draft.

Diego Sotero de Barros Pinangé conceived and designed the experiments, performed the experiments, analyzed the data, prepared figures and/or tables, authored or reviewed drafts of the article, and approved the final draft.

Raquel Moura Machado conceived and designed the experiments, performed the experiments, analyzed the data, prepared figures and/or tables, and approved the final draft.

Santelmo Vasconcelos conceived and designed the experiments, analyzed the data, prepared figures and/or tables, authored or reviewed drafts of the article, and approved the final draft.

Bruno Sampaio Amorim conceived and designed the experiments, analyzed the data, prepared figures and/or tables, and approved the final draft.

Charles Roland Clement conceived and designed the experiments, authored or reviewed drafts of the article, and approved the final draft.

The following information was supplied regarding data availability:

The synthesized data and the original contributions are available in the article and the Supplemental File.

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
