# Peer review of "Relationship between fruit phenotypes and domestication in hexaploid populations of biribá (Annona mucosa) in Brazilian Amazonia"

_PeerJ, doi:10.7717/peerj.14659_

## Round 0.1 · original submission · Minor Revisions

Dear Authors

Kindly carefully see the reviewers' all suggestions one-by-one or step by steps (see carefully all comments given by all four reviewers), include their suggestions and re-submit your revised manuscript. Thank you.

Editor

Reviewer 1 ·

Basic reporting

The paper could be published it is well written and the findings and information presented is interesting.

Experimental design

There is no experimental design as such but the sampling is enough to provide information about the species morphology, it will be interesting to include materials from other countries to further elaborate on the species.

Validity of the findings

The findings are novel, the conclusions correspond well to the level of research

Reviewer 2 ·

Basic reporting

no comment.

Experimental design

This study fills an identified knowledge gap that previous studies on the cytogenetic and genome size of A. mucosa lacking of information about the phenotype of the fruits and absenting of exact collection location.
Although the methods described with sufficient details in this study, the sample size in phylogenetic analyses is limited (only bout 30% species in this genus were sampled), the divergence time of the species remains to be further estimated.

Validity of the findings

no comment.

Additional comments

more discussion should be added to explain the possible impact of limited sampling in phylogenetic analyses on the result of divergence time of the studied species. (because the conclusion of " polyploidization events occurred before the domestication of the species" depends on the estimated divergence time).

Reviewer 3 ·

Basic reporting

The manuscript by Serbin et al. performed chromosome counts of A. mucosa from central and western Brazilian Amazonia, and estimated genome size by flow cytometry. Phylogenetic reconstruction with publicly available data using a Bayesian framework were then performed. And they concluded that A. mucosa was already polyploid and human selection is the main evolutionary force behind fruit size and fruit morphological variation in Annona mucosa. This study is interesting as concerning the evolution of the A. mucosa. However, before accepted for publication, several concerns should be addressed as follow:
1.The results section is too simple. For example, in Line 252 “The reconstruction using ChromEvol revealed x = 7 as the ancestral chromosome number for the genus nnona (Fig. 3a), as did the reconstruction using the R package Phytools (Fig. 3b)”. There should be partial interpretation and explanation for Fig.3.
2.In line 321, The author conclude that the ample phenotypic variation in fruit characteristics, including weight and differences in carpel protrusions, is not associated with an increase in chromosome number or with different genome sizes in Annona mucosa, suggesting that human selection is the main evolutionary force behind fruit size and morphological variation. However, the results did not show that. More details should be supplemented in the results section for this conclusion.
3.In the methods part, the author mentioned that “Subsequently, the sequences of markers for each of the species were concatenated for analysis, as done by Guo et al. (2017) in the major phylogeny for the Annonaceae. ” How many markers were collected for each species, and where are these markers distributed, I suggested these results be presented in the results part, since the density and distribution of these markers are of great influence to the analyses.
4.The format of the references should be carefully checked.

Experimental design

How many markers were applied for the phylogentic analyses will directly affect the results. The author did not show these analyes and results.

Validity of the findings

The conclusion shoud be well establised.

Reviewer 4 ·

Basic reporting

The basic reporting of this article is adequate. The English, for the most part, is descent, but I have made recommendations where appropriate.

A sufficient field background is provided and professional article structure is given to a descent level. Again, I have made suggestions where needed.

One hypothesis needs to be revisited - please see additional comments.

Experimental design

For the most part, the experimental design is descent. I have made recommendations in cases where more details in the Methods are needed.

Validity of the findings

Yes, I thought the validity of the findings was appropriate and the benefit to the literature is clearly stated. Data have been provided where needed and conclusions are well stated and linked to the research questions and results.

Additional comments

I think that this article is suitable for publication in this journal. Taking into account some recommended improvements, I think that the science is sound although the results aren't ground-breaking.
* * *
Comments
* * *
Relationships between fruit phenotypes and domestication in hexaploid populations of biriba (Annona mucosa) in Brazilian Amazonia

In this paper, Serbin et al. examine whether variation in the size of biriba fruits is associated with different levels of ploidy, whether the morphological variation of the carpel protrusion on biriba fruits is associated with genome size, and whether polyploidization events occurred before the domestication of this species. I enjoyed reading this manuscript and learning about one of the interesting fruits of the Amazonia that I may well have seen in the past. My general comments pertain to the lack of justification of one of the hypotheses and how you present the results of your models. I am curious to know why you think there would be an association between carpel protrusions and genome size? For me, I can see the link between single-celled traits (e.g. stomata size) and genome size, as larger genomes require larger cells, but it does not logically follow that this would be related to carpel protrusions.

Also, as mentioned a couple of times in the 'Specific comments', I suggest being cautious about how you word your modelling results. My view is that there no model that is 100% correct, and our scientific language must reflect that.

I am also left wondering whether a larger geographical sampling scale might reveal other patterns in the association among genome size, ploidy level and fruit morphology for this species. Could this be addressed? Perhaps there was a geographic sampling bias that affected your results? Since you basically have a null results, there might not be too much to say about this in the discussion.

Other than those comments, most of my other comments are detail-orientated. However, I do recommend spending a little more time wording some of your ideas, as I found them a little difficult to read at times.



Specific comments

**Introduction**

I appreciate that the study objectives are clearly stated at the end of the Introduction.

I notice that your first objective is appropriately justified, but I do not see any justifications explaining why one would expect the sizes of the carpel protrusions on the fruits to be associated with genome size. Do you have some information that you could add to justify this hypothesis?

I think that the overall organization of the Introduction is reasonable for this journal. You discuss the link between domestication and polyploidy before introducing the family and study species. The Introduction then continues to explain the morphological variation associated with human selection and polyploidy in the species, the current knowledge of its cytology, the frequency of polyploidy in the different clades and the introduction of your study objectives.

Lines 48-49: I don't think that this sentence is needed. You can start the next sentence is a manner like: "Domestication results in a set of modified traits"

Lines 50-51: I suggest adding a little more detail about the domestication syndrome here, because not all of readers will know exactly what it entails.

Lines 58-60: I recommend rewording this sentence because the use of commas makes it difficult to follow.

Lines 70: Since we can't know exactly which year different lineages diverged because of incomplete fossil records, incomplete sampling of existing taxa, imperfect modelling approaches, I recommend being careful how you state divergence times. For example, you can say that recent modelling shows that the group diverged 52.5 Mya, instead of stating that is did diverge 52.5 Mya. There are several ways to state this, but my overall suggestion is make sure to indicate that this is only an estimate, not a sure fact.

Line 83: I suggest changing the word expressive. It reads a little odd in this usage.

Line 88: Consider using a comma before both so that there aren't two consecutive 'or's in the sentence.

Line 93: I also recommend considering to reword this sentence so that it connects better with the ideas before and after, because it is not clear why you are mentioning 'Brassica' as the only clade to show ample morphological variation corresponding with various ploidy levels.

Line 103: The second 'different' is redundant here.

Line 103: You are missing a comma after "A. mucosa".


Table S1 caption: I recommend rewriting the caption as it is not exactly clear what you mean. It took me a bit to realize that you are only presenting those species with existing cytological data. Also - please capitalize 'Annona' in the caption title.


**Materials & Methods**

Fig. 1 - I recommend making the outlines of the Brazil map darker. Also, could you make the three colours representing the sampling regions more divergent, as I am having problems trying to distinguish them.


Table 1 - the column labels require readjustment, as there seems to have been a shift.

Line 127: No capital is needed for smooth fruits.

Line 140: I recommend splitting the sentence at 'were'.

Line 158: Did you use a nonparametric ANOVA because your data were non-normal or there was heteroscadescity in the residuals? I think that is worth mentioning that here.

Line 173: What settings did you use in jModelTest?

Lines 176: Did you use any specific settings for MrBayes, or was the analysis run using default settings? If it was the default settings, what were they? You can mention here that you were conducting a partitioned analysis with substitution models specified to the DNA regions specified in Table T3.

Lines 194-197: This sentence needs to be rewritten to improve the english.

I am a little confused as to why it was needed to repeat the analysis using two different bayesian approaches. I see that the BEAST analysis was only using the subset of species with chromosomal information (Line 182), but I think that this merits justification as it is not a standard approach from my experience.

LInes 199-202: Was any other additional curation done on this data before you used it? Based on experience, our lab group has found several instances of pseudoreplication and the inclusion of unreliable outliers in the datasets. However, this unreliability varies per clade.

Table S4: The section of the caption that reads "ChromEvol software for phylogeny." sounds odd. Is there a way to reword it.

Lines 209-210: I would reword this to: "In addition to AIC estimates,""


Figure 3: - It is a little hard to read the main part of the figure. Is it possible to make the coloured circles thinner, while making the numbers inside of them larger and more prominent?
- after semicolons, it is not needed to use a capital.
- I am not sure what 'later mean ages means'. Is it possible to reword this to make it more clear?

Lines 214-215 - I am not sure what you mean here. Can you reword it?

Figure 2 - I am confused with the number of the subfigures, as they don't match with the caption. I think that this needs to be rewritten.
- how did you select which chromosomes slide to show for each fruit type - was it randomly selected? I would think that there would be a variation in metaphases within each fruit type.
- also, I am not sure why one would expect there to be a difference in how metaphases stain for each fruit type? Why is this important? I see that you mention in the results (Lines 232-233) that there was no discernible relationship - why would one expect otherwise.

Please provide captions for the supplementary figures.


**Results**

Line 240: Change "used" to "studied/examined/investigated"

Line 244: Change "with" to "for".

Lines 246-250: I suggest changing this to clarify that these are only model estimates and not actual dates. For example, "The BEAST analysis estimated the divergence of genus XXX to be..."

Line 252: Be careful with inconsistencies in formating (e.g. Fig. 3a/3b).


**Discussion**

Line 260: Do you mean chromosome number here or ploidy level?

Lines 261-262: I would word this to show that it is the result of modelling, not absolute fact.

Lines 265-266: I find that this sentences is oddly worded. Can you reword it?

Line 267: I think that it would be better to use ploidy level here instead of chromosome number.

Line 276: I am a little confused here. I think that it might be better two state that the wild species have variable ploidy levels. You can then follow that you that there is no influence of selection on ploidy level in 'Annona'. The way that it is worded currently implies that there could be selection for smaller genomes in domesticated species.

Lines 278-285: Is there a simpler word than 'provenance' to use? Do you mean origin? I am not used to seeing this word in scientific formal text.

Lines 286-293: Do you have any speculations about why there is no association between genome size and ploidy level in 'A. mucosa'? Also, why would the heterchromatic regions have different banding? What is the significance of this? I think that there is room to expand on ideas such as this a little more here.

Lines 296-298: I find this to be oddly worded. Could you rewrite it to make it read a little more fluently?

LInes 304-305: Reword so that it is reads a little more clearly.

Line 326: I suggest changing this to read: "and was also inferred to be polyploid" (put the adverb between the helping word and the verb in this case).

Line 454: Capitalize 'Annals of Botany'

---

## Round 0.2 · accepted · Accept

The authors have addressed all of the reviewers' comments. The manuscript is now acceptable for publication.